Article 

# Broadened quantum critical ground state in a disordered superconducting thin film

Koichiro Ienaga [1] ✉, Yutaka Tamoto[1], Masahiro Yoda[1], Yuki Yoshimura[1], Takahiro Ishigami[1] & Satoshi Okuma [1]

A superconductor-insulator transition (SIT) in two dimensions is a prototypical quantum phase transition (QPT) with a clear quantum critical point (QCP) at zero temperature ($T = 0$). The SIT is induced by a field $B$ and observed in disordered thin films. In some of weakly disordered or crystalline thin films, however, an anomalous metallic (AM) ground state emerges over a wide $B$ range between the superconducting and insulating phases. It remains a fundamental open question how the QPT picture of the SIT is modified when the AM state appears. Here we present measurements of the Nernst effect $N$, which has great sensitivity to the fluctuations of the superconducting order parameter. From a thorough contour map of $N$ in the $B$-$T$ plane, we found a thermal-to-quantum crossover line of the superconducting fluctuations, a so-called ghost-temperature line associated with the QPT, as well as a ghost-field line associated with a thermal transition. The QCP is identified as a $T = 0$ intercept of the ghost-temperature line inside the AM state, which verifies that the AM state is a broadened critical state of the SIT.

A quantum phase transition (QPT) between competing ground states at zero temperature ($T = 0$) is a central paradigm seen in a variety of systems such as magnetic materials, cold atom gases, and two dimensional (2D) electrons[1–3]. A prototypical example is a field($B$)-induced superconductor-insulator transition (SIT) observed in disordered 2D superconductors, which shows a clear quantum critical point (QCP) of $(2 + 1)$D quantum systems[4–6]. Since the localization theory prohibits a metallic ground state in 2D systems[7], quantum fluctuations drive a direct transition between condensation and localization of electrons, leading to metallic dissipation with a saturated resistance only at the QCP.

However, an anomalous metallic (AM) ground state emerges unexpectedly over a wide $B$ range in weakly disordered amorphous films[8–11], crystalline 2D superconductors[12–14], and Josephson junction arrays (JJAs)[15–17]. The origin of the superconductor-metal-insulator transition (SMIT) has been debated[8–22] mainly from the following two viewpoints. One is a mechanism of the metallic dissipation at $T = 0$. Some of theories predicted that it is attributed to quantum creep of vortices[18–20], and recent experiments have detected its convincing evidence[10,15,16]. The more fundamental viewpoint is how the QPT picture is modified in the SMIT. A plausible explanation is that the AM state originates from broadening of the SIT[10,19,20]. An existence of such a broadened critical ground state, a so-called quantum critical phase, has attracted much attention in heavy fermion compounds and organic magnets[23,24]. To verify this scenario, the QCP must be found inside the broadened critical state. However, clear experimental evidence is still lacking because the critical scaling analysis for resistance data that is frequently used to determine the QCP of the SIT fails for the SMIT[9]. Furthermore, there are other possibilities including a two-step transition with two QCPs[25], and a single QCP at the metal-insulator transition point[17]. Therefore, it is indispensable to detect the exact location of the QCP using techniques applicable to 2D films beyond the resistance measurements.

A direct way to detect the QCP is to reveal the quantum critical fluctuations of an order parameter around the QCP. A correlation length of the quantum fluctuations diverges at the QCP as a function of a non-thermal parameter such as $B$[1–3], just like a correlation length of the thermal fluctuations diverges at a thermal transition temperature

[1]Department of Physics, Tokyo Institute of Technology, 2-12-1 Ohokayama, Meguro-ku, Tokyo 152-8551, Japan. ✉e-mail: ienaga.k.aa@m.titech.ac.jp; koichiro.ienaga@gmail.com

as a function of $T$. Two types of fluctuations exist in the superconducting order parameter $\Psi = |\Psi| \exp(i\theta)$. The fluctuations of the amplitude $|\Psi|$ arise from short-lived Cooper pairs that persist deep inside the normal state, and the fluctuations of the phase $\theta$ originate from mobile vortices in a vortex-liquid state. In previous reports for 3D superconductors, the thermal critical behavior of the amplitude fluctuations was detected clearly through the Nernst effect above a mean-field transition temperature $T_{c0}$, which leads to an exact determination of $T_{c0}$[26-33]. These studies utilized a high sensitivity of the Nernst effect to the fluctuations. The Nernst signals generated by the superconducting fluctuations are much larger than the signals from normal electrons as reported in many experimental[10,24,26-38] and theoretical works[39-45]. This is in contrast to the resistivity and magnetic susceptibility, where a weak contribution of the fluctuations is buried in the strong intensity from normal electron scattering and the Pauli susceptibility, respectively.

In this work, we have applied the measurements of the Nernst effect to a 2D film of amorphous (a-)$Mo_xGe_{1-x}$ with a thickness of 10 nm, which shows the $B$-induced SMIT[8-10], down to 0.1 K in the quantum regime. By revealing an evolution of the quantum fluctuations across the SMIT, we have found that the QCP is located inside the AM state. The result verifies the view that the AM state is caused by a broadening of the SIT. This work also shows that the Nernst effect is a very useful method to detect a QCP in superconducting systems as reported for the SIT[38] as well as in strongly correlated systems[37,46].

## Results

Figure 1a, b show the $T$-dependent sheet resistance $R_\square(T)$ in different $B$. $R_\square(T)$ in $B = 0$ was measured as a function of $T$, while $R_\square(T)$ in $B \neq 0$ was converted from the magnetoresistance (MR) in Fig. 1e and

Supplementary Fig. S1. A mean-field transition temperature $T_{c0}(\equiv T_{c0,R})$ and a zero-resistance temperature $T_c$ in $B = 0$ are 2.36 and 1.40 K, respectively (see Methods). With increasing $B$, the transition curve shifts to the low-$T$ region. We plot $T_c(B)(\equiv B_c(T))$ in a $B$-$T$ plane (Fig. 2a–c) with solid black circles (see Supplementary Fig. S1). $B_c(T)$ corresponds to a boundary between the vortex-glass phase with zero resistance and the vortex-liquid phase with nonzero resistance. By linearly extrapolating $B_c(T)$ to $T = 0$, $B_c(0)$ is estimated to be 2.5 T.

Above $B_c(0)$, $R_\square(T \to 0)$ saturates to nonzero resistance lower than $R_\square(3.0\,\mathrm{K})$ ($\equiv R_{\square,n}$) as clearly seen in an Arrhenius plot (Fig. 1c), indicating that $B_c(0)(\equiv B_{SM})$ is a boundary field separating the superconducting phase and the AM state. Dashed straight lines in Fig. 1c represent a thermally activated form $R_\square = R'_\square \exp(-U/k_BT)$, where $R'_\square$ is a coefficient, $U$ an activation energy for vortex motion, and $k_B$ the Boltzmann constant. Deviations from the dashed straight lines are marked with open red circles, which correspond to crossover temperatures $T_{cross}$ from the thermal to the quantum creep regime. They are plotted with crosses in the $B$-$T$ plane in Fig. 2a, b. $U$ extracted from Fig. 1c is plotted as a function of $B$ in Fig. 1d. The data points are well fitted by $U = U_0 \ln(B_0/B)$[8,47] with $U_0/k_B = 2.9$ K and $B_0 = 5.0$ T as indicated by a straight line. The successful fit with the value of $B_0$ close to an upper critical (crossover) field $B_{c2}(0) = 5.5$ T (as shown later) is consistent with the collective creep theory[8,47].

Figure 1e shows an enlarged view of the MR curves at different $T$. The metal-insulator transition field $B_{MI} = 5.56$ T is given as a field at which all of the MR curves cross and $R_\square$ is independent of $T$. Above $B_{MI}$, $R_\square$ shows a logarithmic-like increase with a decrease in $T$ as seen in Fig. 1b. As mentioned in our previous report[10], we regard the weak localization behavior as a character of an insulator[6,10,11], which is expected to cross over to a strongly insulating state with exponential

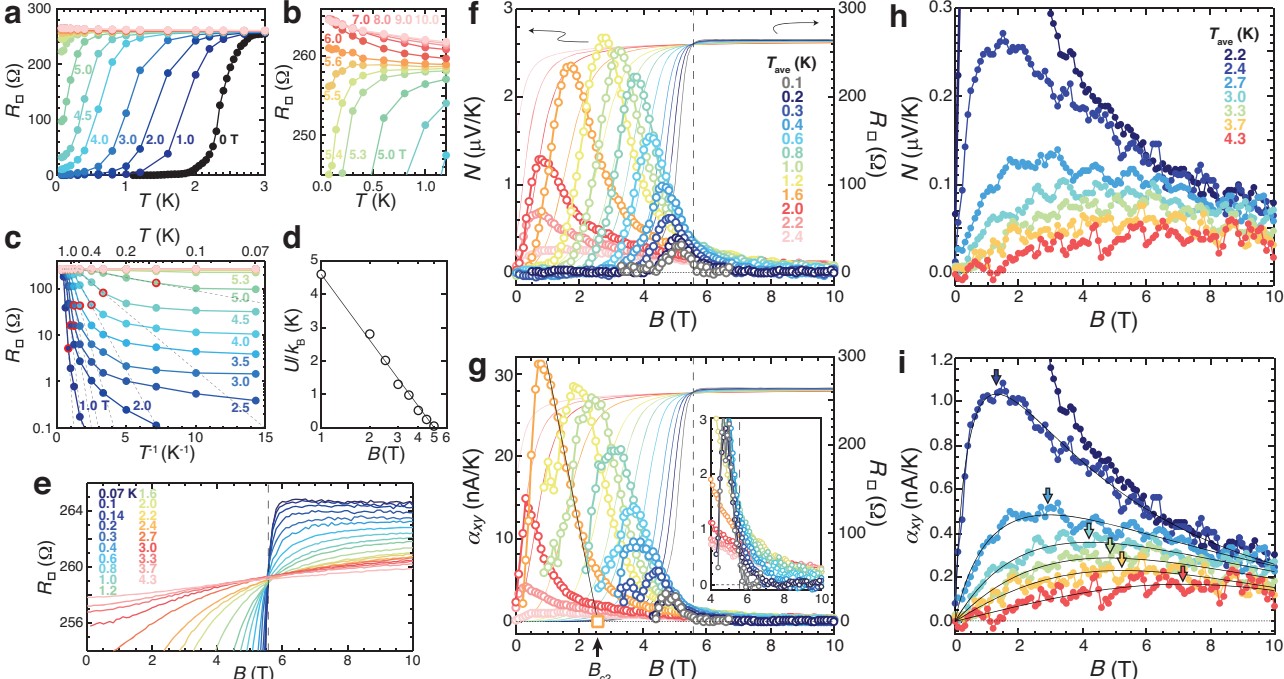

**Fig. 1 | Resistance and Nernst effect in a 2D superconductor exhibiting the field-induced SMIT. a** The $T$ dependence of $R_\square$ in different $B$. $R_\square$ in $B = 0$ was taken as a function of $T$ and $R_\square$ in $B \neq 0$ are plotted from MR in (**e**) and Supplementary Fig. S1. **b** The high-$B$ region of (**a**). **c** An Arrhenius plot of (**a**). Open red circles denote $T_{cross}$ at which $R_\square$ deviates from a thermally activated form drawn by dashed lines. **d** The $B$ dependence of $U$ in the thermally activated form. A black line shows a theoretical fit[8,47]. **e** An enlarged view of MR curves at different $T$. All the MR curves cross at $B_{MI}$ indicated by a dashed line. $N$ (**f**) and $\alpha_{xy}(= N/R_\square)$ (**g**) as a function of $B$ at different $T$

($\leq 2.4$ K). The MR curve at each $T$ is also shown, where the lines with different colors correspond to the data of $N$ and $\alpha_{xy}$ with the same colors. A black straight line in (**g**) shows an example for a linear extrapolation of $\alpha_{xy}$ to determine $B_{c2}$. In the inset of (**g**), $\alpha_{xy}$ in the high-$B$ region is enlarged. Vertical dashed lines indicate $B_{MI}$. The $B$ dependence of $N$ (**h**) and $\alpha_{xy}$ (**i**) at high temperatures ($T \geq 2.2$ K). Black curves in (**i**) are drawn by an ad hoc function to extract $B^*$ as a peak field as indicated by colored arrows (see main text).

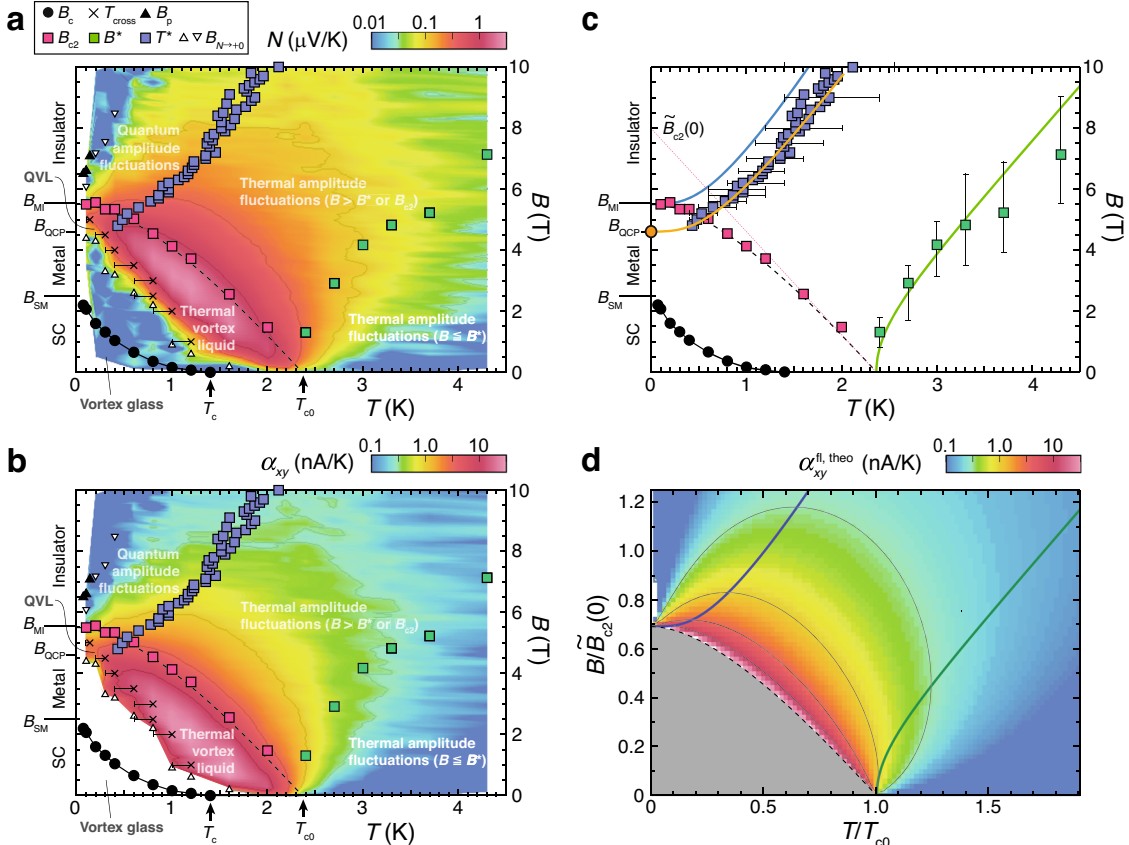

**Fig. 2 | Superconducting fluctuations revealed from the Nernst effect.** Contour maps of $N$ (**a**) and $\alpha_{xy}$ (**b**) constructed from Fig. 1f–i. Black circles, crosses, and black triangles denote $B_c$, $T_{cross}$, and $B_p$ determined from $R_\square$, respectively. White triangles and white inverted triangles represent $B_{N\to+0}$ obtained from $N$. Red, green, and blue squares show $B_{c2}$, $B^*$, and $T^*$, respectively, which are determined from $\alpha_{xy}$. A dashed curve is a fit by WHH theory[52]. **c** The characteristic temperatures and fields obtained experimentally and plotted in (**a**, **b**) are extracted and displayed in the $B$-$T$ plane. Also, theoretically obtained $B^*_{theo}$ and $T^*_{theo}$ extracted from (**d**) are plotted as green and blue lines, respectively. A dotted red line is an extrapolation of the linear GL region of $B_{c2}(T)$. An orange line is a fit to $T^*$ by vertically shrinking the $T^*_{theo}$ line

($\equiv T^*_{theo,m}$). A solid orange circle indicates $B_{QCP}$. Error bars for $B^*$ and $T^*$ represent ranges of $B$ and $T$ in which $\alpha_{xy}(B)$ at fixed $T$ and $\alpha_{xy}(T)$ at fixed $B$ exceed 95 % of their peak amplitudes, respectively. **d** A theoretical result of $\alpha^{fl}_{xy}(T,B)(\equiv \alpha^{fl,theo}_{xy}(T,B))$ calculated for a 2D superconductor based on the Gaussian fluctuations[43,44] using the codes provided in ref. 43. Note that the input parameters are only $T_{c0} = 2.36$ K and $B_{c2}(0) = 5.5$ T obtained experimentally. The $T$ and $B$ axes are normalized by $T_{c0}$ and $\tilde{B}_{c2}(0)$, respectively. Green and blue lines represent the theoretically obtained $B^*(\equiv B^*_{theo})$ and $T^*(\equiv T^*_{theo})$ respectively. A blank area near $T = 0$ indicates negative $\alpha_{xy}$ due to strong quantum fluctuations[44].

divergence at much lower temperatures[5,48]. The weakly localized state is also identified as a dirty metal caused by quantum correction[5]. From these results, the $B$-induced SMIT is confirmed in this sample.

Figure 1f displays a Nernst signal $N$ measured as a function of $B$ at different $T$ from 0.1 K to 2.4 K. Figure 1g shows the transverse thermoelectric conductivity $\alpha_{xy}$ converted from the relation $N = R_\square \alpha_{xy}$ (see Methods). For comparison, the MR curve at each $T$ is also shown. The $B$ and $T$ dependences of $N$ and $\alpha_{xy}$ are similar to those reported in vortex systems[10,26,29,33–38]. With increasing $B$, $N$ together with $R_\square$ starts to grow above $B_c$. Then, $N$ decreases after showing a peak, which is attributed to a decrease in $\alpha_{xy}$ (Fig. 1g). The maximum amplitudes of the vortex Nernst signal $N_{max}$ in the $B$-$T$ range studied is 2.7 μV/K at 1.2 K in 2.8 T. This value is comparable to $N_{max} = 2.9$ μV/K of the 12 nm-thick a-$Mo_xGe_{1-x}$ film with $T_{c0} = 2.58$ K used in our previous work[10] but smaller than $N_{max} = 8.3$ μV/K of a multi-layered a-$Mo_xGe_{1-x}$ film with $T_{c0} \sim 6$ K[33]. The difference of $N_{max}$ may be due to the difference of effective thickness or dimensionality. In Fig. 2a, b, we plot sensitivity limits of the $N$ signals as $B_{N\to+0}$ with open triangles and open inverse triangles in the low and high-$B$ regions, respectively.

$\alpha_{xy}$ consists of three contributions: $\alpha^\phi_{xy}$ from the mobile vortices in the vortex-liquid state, $\alpha^{fl}_{xy}$ from the amplitude fluctuations, and $\alpha^n_{xy}$ from normal electrons[26]. $\alpha^\phi_{xy}$ is given by $\alpha^\phi_{xy} = s_\phi/\phi_0$[26], where $s_\phi$ is the transport entropy in the vortex core and $\phi_0 (\equiv h/2e)$ the flux quantum.

According to the theory[49], with increasing $B$ near a crossover field $B_{c2}$ in the vortex liquid phase, $s_\phi$ decreases linearly as $s_\phi \propto B_{c2} - B$ and vanishes at $B_{c2}$. Therefore, one can determine $B_{c2}$ and distinguish $\alpha^\phi_{xy}$ in $B \le B_{c2}$ from the other contributions in $B > B_{c2}$ by linearly extrapolating $\alpha_{xy}$ to zero as indicated by a solid straight line in Fig. 1g[10,35]. As seen in the high-$B$ region (Fig. 1g inset), the contribution of $\alpha^n_{xy}$, which increases in proportion to $B$, is negligible in amorphous samples due to a very short mean free path of quasiparticles[10,27–29,38]. Thus, $\alpha_{xy}$ in $B > B_{c2}$ corresponds to $\alpha^{fl}_{xy}$.

Recent studies of the vortex Nernst effect have suggested that a maximum value of the vortex transport entropy per unit layer $s^{sheet}_\phi$ is of the order of $k_B$ in many superconductors[33,50,51]. In our experiment, a maximum amplitude of $\alpha^\phi_{xy} = 30$ nA/K at 1.6 K in 0.8 T is converted into $s_\phi = \phi_0 \alpha^\phi_{xy} = 4.5k_B$. Since $s_\phi$ represents the vortex transport entropy per film thickness (= 10 nm), $s^{sheet}_\phi$ is calculated to be 0.14 $k_B$ supposing a unit layer thickness of ~ 3 Å for a-$Mo_xGe_{1-x}$. This value is still close to $k_B$.

We plot $B_{c2}(T)$ thus obtained with solid red squares in the $B$-$T$ plane (Fig. 2a–c). Near $T_{c0,R}$, $B_{c2}(T)$ decreases almost linearly as expected in the Ginzburg-Landau (GL) theory and seems to connect to $T_{c0,R}$. With decreasing $T$, $B_{c2}(T)$ shows a downward deviation from the linear GL line and saturates to $B_{c2}(T \to 0) = 5.5$ T. The value of $B_{c2}(0) = 5.5$ T coincides with $B_{MI} = 5.56$ T within errors, indicating that

the insulating phase in this sample corresponds to a Fermi insulator without vortices[4,5]. $B_{c2}(T)$ thus obtained is well reproduced by the Werthamer-Helfand-Hohenberg (WHH) theory[52] as shown with a dashed line, where the theoretical value of $B_{c2}(0)$ is given by $B_{c2}(0) = 0.69\widetilde{B}_{c2}(0)$ using $\widetilde{B}_{c2}(0)$ $(\equiv T_{c0}|dB_{c2}/dT|_{T=T_{c0}}) = 8.0$ T obtained from a linear extrapolation of $B_{c2}(T)$ in the GL regime to $T = 0$ (Fig. 2c).

Figure 1h, i show the $B$ dependence of $N$ and $\alpha_{xy}$, respectively, at different $T$ above $T_{c0,R}$. Both show a nonmonotonic behavior with a peak at a certain field $B^*(T)$. The peak heights are more than one order of magnitude smaller than the ones below $T_{c0,R}$ (Fig. 1f, g). As $T$ increases above $T_{c0,R}$, $B^*$ shifts to higher $B$ with a broadening of the peak and a decrease in its height. These behaviors have been observed as common features of $\alpha_{xy}^{fl}$ above $T_{c0,R}$[26-32].

According to the theory based on the Gaussian fluctuations[40-44], $\alpha_{xy}^{fl}$ above $T_{c0,R}$ in $B \ll B^*$ follows the expression $\alpha_{xy}^{fl} = (k_B e^2/6\pi\hbar^2)\xi_{GL}^2 B$, where $\hbar$ is the reduced Planck constant, $\xi_{GL}(T) = \xi_0/\sqrt{\ln(T/T_{c0})}$ the GL coherence length, and $\xi_0$ the Bardeen-Cooper-Schrieffer (BCS) coherence length. This means that the initial slope of $\alpha_{xy}^{fl}(B)$, namely $\alpha_{xy}^{fl}/B|_{B\to 0}$, is a measure of $\xi_{GL}(T)$, which is the correlation length of the amplitude fluctuations below $B^*$. Thus, a temperature at which $\alpha_{xy}^{fl}/B|_{B\to 0}$ diverges corresponds to $T_{c0}$ defined from the divergence of $\xi_{GL}(T)$. By plotting the $T$ dependence of $\alpha_{xy}^{fl}/B|_{B\to 0}$ obtained above $T_{c0,R}$, we can determine $T_{c0} = 2.36$ K (see Supplementary Section III-1), which coincides with $T_{c0,R} = 2.36$ K. With increasing $B$ at a given $T$, the correlation length of the amplitude fluctuations is reduced from constant $\xi_{GL}(T)$ to a cyclotron radius (magnetic length) $l_B(B) = \sqrt{\hbar/2eB}$ when $l_B(B) \le \xi_{GL}(T)$[26,28-32]. Consequently, $\alpha_{xy}$ exhibits a peak around $B = \phi_0/2\pi\xi_{GL}^2 (\equiv B^*)$, which is called a ghost (critical) field[26,28-32,44,53]. This means that a temperature at which $B^*(T) \to 0$ with the divergence of $\xi_{GL}(T)$ also corresponds to $T_{c0}$. To determine $B^*(T)$, we fit $\alpha_{xy}(B)$ in Fig. 1i with an ad hoc fitting function $pB\exp(-qB^r)$ (solid curves)[38], where $p$, $q$, and $r$ are positive fitting parameters. $B^*(T)$ thus obtained is indicated with colored arrows in Fig. 1i and plotted in Fig. 2a–c with solid green squares. $B^*(T)$ approaches zero toward $T_{c0}$. The existence of $B^*(T)$ in the normal state is also confirmed by scaling analysis for $\alpha_{xy}/B$ (see Supplementary Section III-2).

Figure 2a, b show the contour maps of $N(T,B)$ and $\alpha_{xy}(T,B)$, respectively. Below the $B_{c2}(T)$ line in the thermal vortex-liquid phase, $N$ and $\alpha_{xy}$ have large magnitudes, which are due to vortex motion. They are observed down to the lowest temperature 0.1 K in the high-$B$ range of the AM state, indicating that the AM state originates from the vortex liquid due to the quantum fluctuations, namely quantum vortex liquid (QVL), as reported in our previous work[10]. Meanwhile, above the $B_{c2}(T)$ line, the contribution of the amplitude fluctuations seen from $N$ and $\alpha_{xy}$ is relatively weak but persists in a wide range of the $B$-$T$ plane with arc-shaped contour lines (light gray lines).

In the main panel of Fig. 3, the $T$ dependence of $\alpha_{xy}(T)$ is plotted for different $B$ from 4.8 T to 10 T. In the inset of Fig. 3, $\alpha_{xy}(T)$ above $B_{c2}(0)$ is enlarged and shown. $\alpha_{xy}(T)$ shows a peak at a certain temperature $T^*(B)$. With an increase in $B$, the peak becomes broad, its height decreases, and $T^*$ increases. These behaviors closely resemble those of theoretically obtained $\alpha_{xy}^{fl}(T)$ above $B_{c2}(0)$ (Fig. 2b in ref. 44), where the peak temperature $T^*(B)$ called a ghost temperature separates the thermal and quantum regimes of the amplitude fluctuations. A similar thermal-to-quantum crossover temperature has been theoretically predicted[41-43,54]. Although a seemingly similar behavior was observed earlier in the high-$B$ region of cuprates[36], its origin was different. It was interpreted as originating from vortices with large quantum fluctuations[36,39].

According to the theory[43], the quantum amplitude fluctuations below $T^*(B)$ are described as puddle-like fluctuations with a size of the quantum correlation length $\xi_{qf}(B) = \xi_0/\sqrt{\ln(B/B_{QCP})}$, which diverges at a QCP denoted by $B_{QCP}$ (see Supplementary Section III-3). In our MR data (Fig. 1e), indeed, the existence of the quantum amplitude

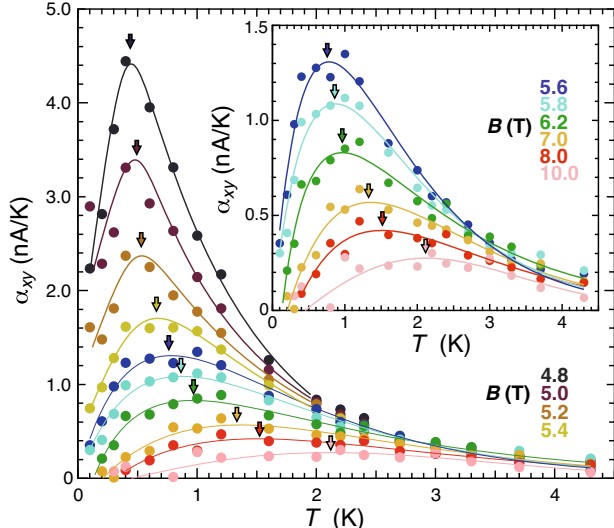

**Fig. 3 | Estimation of a ghost temperature $T^*$.** The $T$ dependences of $\alpha_{xy}(T)$ for different $B$ from 4.8 T to 10 T. In the inset, $\alpha_{xy}(T)$ above $B_{c2}(0)$ is enlarged. The ghost temperature $T^*(B)$ is estimated from a peak temperature as indicated by colored arrows. It is obtained from the fit lines using an ad hoc function and smoothly connected eye guide lines for the data above and below $B_{c2}(0)$, respectively, both of which are shown with solid colored curves.

fluctuations is suggested by a slight hump around $B_p \sim 6.5–7.0$ T ($> B_{c2}(0)$) near $T \approx 0$ ($T = 0.07–0.14$ K)[5,43], as found in our previous report[10]. Furthermore, the theories[41-44] show that $\alpha_{xy}^{fl}(T)$ above $B_{c2}(0)$ at $T \ll T^*$ is proportional to $\xi_{qf}^2 T$ (see Supplementary Section III-3). This means that an initial slope of $\alpha_{xy}^{fl}(T)$, namely $\alpha_{xy}^{fl}/T|_{T\to 0}$, at given $B$ is a measure of $\xi_{qf}(B)$. Therefore, a field at which $\alpha_{xy}^{fl}/T|_{T\to 0}$ diverges with $T^*(B) \to 0$ corresponds to a QCP determined from the amplitude fluctuations. To determine $T^*$, we fit $\alpha_{xy}(T)$ above $B_{c2}(0)$ with a function $p(T - T_0)\exp(-qT^r)$ (colored solid lines) similar to the one mentioned above, where $T_0$ corresponds to the sensitivity limit of $\alpha_{xy}$ at $B_{N\to 0}$. For $\alpha_{xy}(T)$ below $B_{c2}(0)$, we draw eye-guide lines (colored solid lines). $T^*(B)$ thus obtained is indicated with colored arrows in the main panel and the inset of Fig. 3 and plotted in Fig. 2a–c with solid blue squares. With decreasing $B$, $T^*(B)$ approaches zero toward a field below $B_{c2}(0)$. This indicates that the location of $B_{QCP}$ determined from the amplitude fluctuations is inside the AM state. Thus, $B_{MI}(\approx B_{c2}(0))$ is not a QCP but a crossover point, below which the quantum amplitude fluctuations are transformed into the QVL. To our knowledge, this is the first experiment to determine the ghost temperature $T^*(B)$ in the $B$-$T$ plane and the QCP inside the AM state. The existence of $T^*(B)$ is also confirmed by scaling analysis for $\alpha_{xy}/T$ (see Supplementary Section III-4).

Figure 2d shows the contour map of the theoretical values of $\alpha_{xy}^{fl}(T,B)(\equiv \alpha_{xy}^{fl,theo}(T,B))$ calculated for a 2D superconductor based on the Gaussian fluctuations[42-44] using the codes provided in ref. 43. Here, we input only two parameters, $T_{c0} = 2.36$ K and $B_{c2}(0) = 5.5$ T obtained experimentally. The $T$ and $B$ axes are normalized by $T_{c0}$ and $\widetilde{B}_{c2}(0)$, respectively. A theoretical line of $B_{c2}(T)$ (a dashed line), which is drawn by the WHH theory[52], is identical to the fitting line of the experimental data of $B_{c2}(T)$ (a dashed line in Fig. 2a–c). It is seen from the contour maps that the theoretical values of $\alpha_{xy}^{fl,theo}(T,B)$ in Fig. 2d reproduce almost quantitatively the experimental data of $\alpha_{xy}^{fl}(T,B)$ in Fig. 2b, both of which persist in the wide range of the $B$-$T$ plane above $B_{c2}(T)$. The green and blue lines in Fig. 2c, d represent the theoretical lines of $B^*(T)(\equiv B_{theo}^*(T))$ and $T^*(B)(\equiv T_{theo}^*(B))$ obtained from the $B$ and $T$ dependences of $\alpha_{xy}^{fl,theo}(T,B)$, respectively. It is found that the experimental data of $B^*(T)$ (green squares) in Fig. 2c are in good agreement with the theoretical values of $B_{theo}^*(T)$ (a green line) within error bars. By contrast, the experimental data of $T^*(B)$ (blue squares) in Fig. 2c

clearly fall on a line below the theoretical line of $T^*_{\text{theo}}(B)$ (a blue line). However, when $T^*_{\text{theo}}(B)$ is vertically compressed by a factor of 4.6/5.5 as shown with an orange line ($\equiv T^*_{\text{theo,m}}(B)$), the experimental data of $T^*(B)$ well fall on the modified theoretical line of $T^*_{\text{theo,m}}(B)$ (see also Supplementary Section II). From the end point of the orange line, $T^*_{\text{theo,m}}(B \to B_{\text{QCP}}) \to 0$, marked with a solid orange circle, $B_{\text{QCP}} = 4.6$ T is determined. To our knowledge, this value is first obtained from the present Nernst-effect measurements that probe the amplitude fluctuations, which could not be obtained from conventional measurements, such as resistance. Furthermore, the criticality at $B_{\text{QCP}}$ is also confirmed from the $B$ dependence of $\alpha^{\text{fl}}_{xy}/T|_{T\to 0}(\propto \xi_{\text{qf}}(B)^2)$ above $B_{\text{c2}}(0)$, which shows a divergent behavior toward $B_{\text{QCP}}$ (see Supplementary Section III-3).

## Discussion

Finally, let us discuss the origin of the AM state. Generally, the quantum critical regime is defined by $\xi_{\text{qf}}(B) > L_\theta(T)$ [1-3], where $L_\theta(T) \sim T^{-1/z}$ is a dephasing length and $z$ is a dynamical critical exponent[1]. In the standard QPTs including the SIT, with decreasing $T$, the field range of the quantum critical regime is reduced to zero toward the QCP as illustrated in Fig. 4a because $L_\theta(T)$ diverges as $T \to 0$. However, if $L_\theta(T)$ is saturated to a constant value below a certain temperature $T_{\text{sat}}$ for some reasons[55], the field range satisfying $\xi_{\text{qf}}(B) > L_\theta(T_{\text{sat}})$ remains nonzero at $T = 0$ as in Fig. 4b[20], resulting in the quantum critical ground state that spans over an extended field range. Indeed, the saturation of $L_\theta(T)$ has been observed in the AM state of a JJA[16]. In our experiment, the existence of the QCP is clearly detected as $B_{\text{QCP}}$ inside the AM state. Moreover, the quantum critical behavior[37,46,56] is observed not only at $B_{\text{QCP}}$ but also in the wide range of $B$ inside the AM state in the present sample (see Supplementary Fig. S4) as well as in our previous report[10]. From these results, we conclude that the AM state originates from the broadening of the SIT. This is consistent with the theoretical prediction[19] that there exists a broadened quantum ground state between $B_{\text{SM}}$ and $B_{\text{MI}}$. It is likely that the thermal-to-quantum crossover (ghost-temperature) line $T^*(B)$ obtained in this work reflects a boundary where $\xi_{\text{qf}}(B) = L_\theta(T)$ above $T_{\text{sat}}$ (see Supplementary Section III-4).

A possible origin of the saturation of $L_\theta(T)$ in superconductors is a coupling between Cooper pairs and a fermionic dissipative bath, which makes the pure SIT unstable[9,15-17,20,57]. Meanwhile, in magnetic materials, the emergence of the quantum critical phase is attributed to magnetic frustration, leading to a possible quantum spin liquid[23,24]. Indeed, ref. 19 theoretically suggests that the critical AM state has characteristics analogous to the quantum spin liquid. Thus, some frustration effect on the Cooper pairs from a dissipative bath can be an origin of the AM state.

To conclude, by constructing the thorough contour map of the Nernst signal in the $B$-$T$ plane, we reveal an evolution of the superconducting fluctuations in the disordered 2D superconductor exhibiting the SMIT. We found the thermal-to-quantum crossover line (ghost-temperature line) $T^*(B)$ associated with the QPT, as well as the ghost-field line $B^*(T)$ associated with a thermal transition. The $T = 0$ QCP determined from $T^*(B \to B_{\text{QCP}}) \to 0$ is located inside the AM state, indicating that the AM state is a broadened critical state of the SIT. Our work also shows the usefulness of the Nernst effect to study QPTs in superconducting systems.

## Methods

### Samples

An a-Mo$_x$Ge$_{1-x}$ film with $x = 0.77$ and a thickness of 10 nm was prepared by rf-sputtering onto a 0.15 mm thick glass substrate. The film thickness is comparable to $\xi_0 \sim 8$ nm estimated from $\xi_0^2 = \phi_0/(2\pi B_{\text{c2}}(0))$ with $B_{\text{c2}}(0) = 5.5$ T. To obtain a homogeneous amorphous film, the substrate was mounted on a water-cooled holder rotating with 240 rpm during sputtering. Ag electrodes were deposited 4.2 mm ($= L$) and 5.6 mm ($= l$) apart for longitudinal $V_x$ and transverse voltage $V_y$, respectively. The film thus prepared was protected by depositing a thin SiO layer. $T_{\text{c0},R} = 2.36$ K and $T_{\text{c}} = 1.40$ K are determined from $R_\square(T_{\text{c0},R}) = 0.5R_{\square,\text{n}} = 0.5R_\square(3.0 \text{ K})$ and the resistance drop below the measurement limit, respectively. $T_{\text{c}} = 1.40$ K is reduced compared with $T_{\text{c}} = 6.2$ K for the 330 nm thick film[58]. The broadened transition curve indicates 2D nature of superconductivity, similarly to the 12 nm thick film with $T_{\text{c}} = 2.13$ K used in our previous work[10].

### Resistivity and Nernst-effect measurements

All measurements were carried out using a sample holder installed in a dilution refrigerator[10]. The magnetic field $B$ was applied perpendicular to the film plane. $R_\square = (V_x/L)/(I_x/l)$ was obtained using standard four-terminal dc and low-frequency ac (19 Hz) lock-in methods with a bias current of $I_x \geq 30$ nA in the ohmic regime.

$N$ was obtained from $N \equiv E_y/\nabla T_x = (V_y/l)/(\Delta T/L)$, where $\Delta T = T_{\text{high}} - T_{\text{low}}$ is a temperature difference. $V_y$ was measured with a nanovoltmeter. To set $\Delta T$, a heat current was applied from a heater on one side of the glass substrate to the other side glued on a heat bath. $\Delta T$ was measured by two RuO$_2$ thermometers, which were thermally isolated from the bath but thermally coupled to the two Ag electrodes of $V_x$ through Cu wires. An average sample temperature was defined by $T_{\text{ave}} = (T_{\text{high}} + T_{\text{low}})/2$. $\Delta T$ was fixed to be 30%, ~20%, and ~10 % of $T_{\text{ave}}$ at 0.1 K, 0.2−0.6 K, and 0.8−4.3 K, respectively, to obtain measurable $V_y$ keeping $\Delta T$ as low as possible. At each measurement step, we switched on and off $\Delta T$ to subtract the background thermoelectric voltage observed at $\Delta T = 0$ from $V_y$ at $\Delta T \neq 0$. To correct misalignment of heat flow, we extracted an antisymmetric contribution of $N$ with respect to magnetic field reversal. $N$ from the superconducting fluctuations is approximated by $N = R_\square \alpha_{xy}$ due to the particle-hole symmetry[40,59]. $\alpha_{xy}$ is a thermodynamic quantity proportional to the transport entropy, while $R_\square$ reflects electron scattering and viscosity of vortices.

We checked an effect of external noise using low-pass RC filters with a cutoff frequency of 100 kHz, which is adequate to eliminate the noise effect[21,22]. Although the filters were inserted between the sample and a nanovoltmeter as well as between the sample and a current bias source at room temperature, the results with and without the filters were almost the same. Note that, in the Nernst-effect measurements, the sample was electrically isolated from the current bias source, which is a main possible origin of external noise[21,22].

## Data availability

The data used in this study are available in the main text and the Supplementary Information. All other data are available from the corresponding author upon request. Source data are provided with this paper.

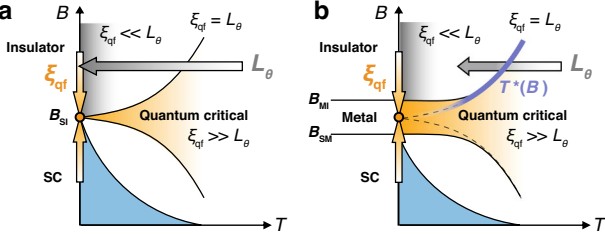

**Fig. 4 | Schematic models of quantum criticality in SIT and SMIT. a** A standard picture of QPTs for the $B$-induced SIT in the $B$-$T$ plane[1-3]. The critical field $B_{\text{SI}}$ of the SIT corresponds to the QCP at $T = 0$. **b** A proposed model of quantum criticality in the SMIT[10,20]. The AM state stems from broadening of the SIT, which is verified in this work by finding $B_{\text{QCP}}$ (solid orange circle) as a field that the experimental $T^*(B)$ line approaches as indicated by a solid blue line. In both (**a**, **b**), the quantum critical regime is defined by the area where $\xi_{\text{qf}}(B) > L_\theta(T)$. Gray and orange arrows indicate the direction of the increase in $L_\theta(T)$ and $\xi_{\text{qf}}(B)$, respectively.

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

## Acknowledgements

We thank R. Ikeda for variable comments, and S. Kaneko and S. Maegochi for fruitful discussions. This work was supported by Grant-in-Aid for Early-Career Scientists (KAKENHI Grant No. 20K14413 to K.I.), Scientific Research(B) (KAKENHI Grant No. 22H01165 to S.O.), and Challenging Research (KAKENHI Grant No. 21K18598 to S.O. and 23K17667 to K.I.) from the Japan Society for the Promotion of Science. K.I. acknowledges support from Tokyo Institute of Technology (Yoshinori Ohsumi Fund for Fundamental Research and ASUNARO Grant).

## Author contributions

K.I. designed the experiments in discussion with S.O. K.I., Y.T., M.Y. and Y.Y. fabricated the sample and conducted measurements. K.I. and T.I analyzed the data. K.I. wrote the manuscript with S.O. All authors discussed the results and commented on the manuscript.

## Competing interests

The authors declare no competing interests.
