## [Peer Review File · Nature Communications]

REVIEWER COMMENTS

Reviewer #1 (Remarks to the Author):

The paper by Ienaga and co-workers reports on a study of Nernst effect in disordered thin films of MoGe focused on the anomalous metallic state appearing between the insulator and the superconductor. The quality of the data is remarkably good, and the analysis is mostly sound. Save for the following points, which need clarification, I do not mind recommending it for publication in Nature Communications:

1. The paper identifies the ghost critical field using the Nernst effect. This was first done in Phys. Rev. B 76, 214504 (2007). The concept of a "ghost critical field" was first coined in J. Phys. C, 18 1305 (1985). Surprisingly, none of these two papers make it to the long list of references.
2. Now, the ghost critical field refers to the existence of a field scale (and therefore a length scale) in the normal state that mirrors the upper critical field (and the coherence length) in the superconducting state. The first claim of this paper is its observation in the normal state of MoGe. This claim is convincing. There is a peak in the field dependence of α_{xy} above T_c and the peak shifts to higher temperature with warming. On the other hand, there is a stronger statement about the ghost critical field, shown in Fig. 1 of PRB 76, 214504 (2007). Plotting N or α_{xy} in the (x,y) plane, where x is the correlation length (extracted from reduced temperature) and y is the magnetic length (extracted from the magnetic field), then the data shows a diagonal symmetry. This is a much stronger statement about the ghost critical field. Do the authors' data support such a strong statement?
3. The main accomplishment of this paper is to have observed for the first time a new temperature scale which shifts upward with increasing magnetic field. The authors attribute this temperature to the equality between the dephasing length and the quantum phase fluctuation length. The experimental observation appears robust, and the interpretation is plausible. However, is there any way to quantify these two length scales at a given temperature and magnetic field? This would lead a strong statement.
4. Comparing the experimental Fig. 2a and theoretical figure 4b, one sees a crucial difference. In the experimental data T^* crosses B_{c2} , B_{MI} and even B_{QCP} . The theoretical line has a margin, which leads to the anomalous metal. Clarification would be helpful.

5. This is not the first study of Nernst effect in MoGe. The vortex Nernst signal was already measured in Phys. Rev. Lett. 126, 077001 (2021), with a peak three times larger than what was seen here. This difference may be due to the difference in the thickness of the thin films studied in the two cases. However, it should not be overlooked.

6. I also note that the vortex α_{xy} peaks to 30 nK/A. Given the amplitude of the quantum of the magnetic flux, this corresponds to a vortex entropy of $\sim 4 k_B$ per layer. I understand that this is not an essential part of the topic discussed in this paper. However, a short discussion would be useful to the reader interested in the observation that vortex entropy per sheet is of the order of the Boltzmann constant in many superconductors (See table 1 in Physica C 591, 1353975 (2021) and table 1 in J. Phys.: Condens. Matter 35 074003(2003)).

Reviewer #2 (Remarks to the Author):

This manuscript describes a comprehensive experimental study of thin MoGe films. The authors use transport and Nernst effect measurements in order to study the electronic properties of the different phases of the layers as a function of temperature and magnetic field. Using detailed analysis of the results the authors identify the anomalous metal regime between the insulator and the superconductor and infer that it is due to a broadening of the well-known and studied "quantum critical point" and the formation of an extended "quantum critical phase".

Though the authors performed very high-quality and detailed experimental and analysis work I do not find this manuscript suitable for publication in Nature Communications for the following reason:

1. My major problem with this manuscript is that, despite the fact that the authors claim to present an SMIT, they do not show any insulating behavior. In Fig. 1 a,b an insulating phase is not seen. The authors claim that above " B_{MI} " the resistance shows a logarithmic-like increase with decreasing temperature (though the exact temperature dependence is not shown). Such a weak T dependence is characteristic of dirty metal behavior due to weak localization (or other phenomena) rather than an insulator that should show exponential dependence of the resistance with decreasing temperature. It is not clear how the authors define the M phase versus the I phase.

2. I am a bit confused about the interplay of quantum amplitude and phase fluctuations. On one hand the authors claim that the AM is due to vortex creep. On passing, I have to say that I find this claim a bit too definitive. The presence of the AM and its possible origin is very much under debate and most of the references quoted by the authors do not conclude voltage creep as the cause for AM. But even if this claim was true this would mean that phase fluctuations and vortices are very important and would dominate the Nernst effect. On the other hand, in their analysis of α_{xy} and determining the limit

for the quantum fluctuation regime the authors only consider amplitude fluctuations. In this respect I find the determination of the quantum fluctuation regime questionable. Since the main point of the manuscript is to determine the broadened quantum critical phase of the AM it is important to identify to identify all sources for fluctuations.

3. Another importance of this work raised by the authors in the conclusion is that the work shows the usefulness of the Nernst effect to study QPTs in superconducting systems. It should be noted that reference 36 showed a detailed study of quantum criticality of the SIT using Nernst effect, though in that case there was no intermediate AM.

4. There are a few small typos in the text for example:

(i) In the middle of page 5 " \hbar " should be the Plank constant and not the Dirac constant.

(ii). Caption of Fig. 1 a rather than "fixed B" it should be "different B".

Reply to Reviewer #1

We would like to thank the reviewer #1 for very positive comments on our work: *'The quality of the data is remarkably good, and the analysis is mostly sound. Save for the following points, which need clarification, I do not mind recommending it for publication in Nature Communications.'* We have revised the paper according to the valuable comments and suggestions.

Comments 1:

1. The paper identifies the ghost critical field using the Nernst effect. This was first done in Phys. Rev. B 76, 214504 (2007). The concept of a "ghost critical field" was first coined in J. Phys. C, 18 1305 (1985). Surprisingly, none of these two papers make it to the long list of references.

Our response:

We appreciate the helpful comment. We add the two important references as Refs. 27 and 53. Although we recognized the importance of PRB 76, 214504 (2007) reported by A. Pourret *et al.*, we instead cited their review article reported later [New J. Phys. 11, 055071 (2009), Ref. 28].

Comments 2:

2. Now, the ghost critical field refers to the existence of a field scale (and therefore a length scale) in the normal state that mirrors the upper critical field (and the coherence length) in the superconducting state. The first claim of this paper is its observation in the normal state of MoGe. This claim is convincing. There is a peak in the field dependence of α_{xy} above T_c and the peak shifts to higher temperature with warming. On the other hand, there is a stronger statement about the ghost critical field, shown in Fig. 1 of PRB 76, 214504 (2007). Plotting N or α_{xy} in the (x,y) plane, where x is the correlation length (extracted from reduced temperature) and y is the magnetic length (extracted from the magnetic field), then the data shows a diagonal symmetry. This is a much stronger statement about the ghost critical field. Do the authors' data support such a strong statement?

Our response:

We sincerely appreciate the helpful suggestion. We add the new section in the supplementary information (section III-2) to provide more convincing evidence of the existence of the ghost critical field $B^*(T)$. The reviewer #1 suggests plotting $\alpha_{xy}(T, B)$ in the (ξ_{GL}, l_B) plane to confirm a diagonal symmetry as reported in PRB 76, 214504 (2007) [Ref. 27]. Although we obtain a result implying the diagonal symmetry, it is difficult to plot our data of $\alpha_{xy}(T, B)$ as a color map in the (ξ_{GL}, l_B) plane. To plot the color map with a constant interval of ξ_{GL} , an interval of T should be narrower as T approaches T_{c0} because $\xi_{GL} \propto (\ln(T/T_{c0}))^{-1/2}$. However, our data of $\alpha_{xy}(B)$ above T_{c0} are measured at 6 temperatures with almost constant intervals (2.4, 2.7, 3.0, 3.3, 3.7, and 4.3 K). Therefore, we instead perform another analysis referring to Ref. 27, which is equivalent to or more convincing than the analysis that the reviewer #1 asked us to do. We plot the data of α_{xy}/B above T_{c0} as a function of B in Fig. S3a. Then, by replotting them as $(\alpha_{xy}/B)/(\alpha_{xy}/B|_{B=0})$ against B/B^* in Fig. S3b, we confirm that all of the data measured at different T collapse on a single curve

as reported in Ref. 27. The successful scaling result is a stronger statement on the existence of $B^*(T)$. Details are explained in the supplementary information section III-2.

On a related note, we revise the method to estimate a value of $\alpha_{xy}/B|_{B=0}$. In the previous manuscript, we estimated the value of $\alpha_{xy}/B|_{B=0}$ from a fitting coefficient p in an *ad hoc* function $pB\exp(-qB^r)$ used in the text. However, we have noticed that the obtained values are slightly overestimated. In the revised version, we determine the value of $\alpha_{xy}/B|_{B=0}$ by plotting α_{xy}/B against B in Fig. S3a, where α_{xy}/B becomes saturated in the low- B region. Consequently, the value $\alpha_{xy}/B|_{B=0}$ becomes slightly smaller in the present version, and the value of T_{c0} deduced from the T dependence of $\alpha_{xy}/B|_{B=0}$ in Fig. S3c changes slightly from 2.38 K in the previous version to 2.36 K in the present version. We revise all relevant part in the main text and the supplementary information.

Comments 3:

3. The main accomplishment of this paper is to have observed for the first time a new temperature scale which shifts upward with increasing magnetic field. The authors attribute this temperature to the equality between the dephasing length and the quantum phase fluctuation length. The experimental observation appears robust, and the interpretation is plausible. However, is there any way to quantify these two length scales at a given temperature and magnetic field? This would lead a strong statement.

Our response:

We again appreciate the helpful suggestion. We add the new section in the supplementary information (section III-4) to provide more convincing evidence of the ghost temperature $T^*(B)$ and to validate the equality between the dephasing length $L_\theta(T)$ and the quantum correlation length $\xi_{\text{qt}}(B)$ at $T^*(B)$. Since it is difficult to quantify $L_\theta(T)$, instead we perform a scaling analysis for α_{xy}/T as performed for α_{xy}/B in the section III-2. By plotting $(\alpha_{xy}/T)/(\alpha_{xy}/T|_{T=0})$ against T/T^* in Fig. S3e, we confirm that all of the data obtained at different B collapse on a single curve. The successful scaling result is a much stronger statement on the existence of $T^*(B)$. From this analysis, furthermore, we find that a characteristic length scale of the amplitude fluctuations changes from $\xi_{\text{qt}}(B)$ below T^* to $L_\theta(T)$ above T^* . Details are explained in the supplementary information section III-4.

On a related note, similar to the case of $\alpha_{xy}/B|_{B=0}$, we revise the method to estimate a value of $\alpha_{xy}/T|_{T=0}$. In the previous manuscript, we estimated the value of $\alpha_{xy}/T|_{T=0}$ from a fitting coefficient p in an *ad hoc* function $pT\exp(-qT)$ used in the text. However, we have noticed the obtained values are slightly overestimated. In the revised version, we determine the value of $\alpha_{xy}/T|_{T=0}$ by plotting α_{xy}/T against T in Fig. S3d, where α_{xy}/T becomes saturated in the low- T region. Furthermore, we also have noticed our mistake in Eq. (2) in the previous version of the supplementary information (= Eq. (5) in the present version). In our analysis, Eq. (6) modified from Eq. (5) should have been used instead of Eq. (5). As a result of these corrections, the B dependence of α_{xy}/T plotted in Fig. S3f is slightly changed but the value of $B_{\text{QPC}} (= 4.6 \text{ T})$ determined from Fig. S3f remains the same.

Comments 4:

4. Comparing the experimental Fig. 2a and theoretical figure 4b, one sees a crucial difference. In the experimental data T^* crosses B_{c2} , B_{MI} and even B_{QCP} . The theoretical line has a margin, which leads to the anomalous metal. Clarification would be helpful.

Our response:

We are concerned that Figs. 2a-c and 4b may have led to confusion. What we conclude is that by experimentally detecting a peak temperature ($= T^*(B)$) from the T dependence of $\alpha_{xy}(T)$, we have revealed the existence of the characteristic line that approaches B_{QCP} as fitted by an orange line in Fig. 2c. In the schematic model for the SMIT (Fig. 4b), the orange fitting line in Fig. 2c corresponds to a dashed line above B_{QCP} that approaches B_{QCP} in the B - T plane. A deviation of a data point of $T^*(B)$ for 4.6 T ($= B_{QCP}$) from the orange fitting line, which concerns the reviewer #1, results from the fact that $T^*(B)$ below B_{QCP} does not have any physical meaning. An analogous deviation has also been known for the ghost critical field $B^*(T)$, which is determined from a peak field in the B dependence of $\alpha_{xy}(B)$. As discussed in Ref. 27, $B^*(T)$ represents the characteristic field where $\zeta_{GL}(T) = l_B(B)$ only above T_{c0} , while it does not have any physical meaning below T_{c0} . Therefore, to avoid confusion, in the revised manuscript we delete the plot of $T^*(B)$ in 4.6 T ($= B_{QCP}$) from Fig. 2a-c and Fig. S2a, and the data of $\alpha_{xy}(T)$ in 4.6 T from Fig. 3.

Furthermore, the reviewer #1 is also concerned about the experimental observation that the experimental $T^*(B)$ line crosses B_{c2} and B_{MI} . However, this behavior is naturally allowed in the condition that $B_{QCP} < B_{c2}$ and B_{MI} because $T^*(B)$ approaches B_{QCP} at $T = 0$. We suppose that the reviewer #1 may have considered that the experimental $T^*(B)$ line coincides with the $\zeta_{qf}(B) = L_\theta(T)$ boundary down to $T \rightarrow 0$, which is represented as the solid line connected to B_{MI} in Fig. 4b. As mentioned above, we interpret the experimental $T^*(B)$ line to be the dashed line connected to B_{QCP} in Fig. 4b. The dashed line is the same as the solid line above T_{sat} but separated from the solid line below T_{sat} . In this sense, we wrote on page 8, line 220 in the previous manuscript (page 8, line 236 in the present manuscript), ‘It is likely that the thermal-to-quantum crossover (ghost-temperature) line $T^*(B)$ obtained in this work reflects a boundary where $\zeta_{qf}(B) = L_\theta(T)$ above T_{sat} .’ To improve clarity, we revise Fig. 4b by clearly showing the correspondence between the dashed line and the experimental $T^*(B)$ line. It is also noted that the relationship between the experimental $T^*(B)$ line and the $\zeta_{qf}(B) = L_\theta(T)$ line is first suggested by our experiment and a detailed theory is not available. Thus, the origin of the deviation between the experimental $T^*(B)$ line and the $\zeta_{qf}(B) = L_\theta(T)$ line below T_{sat} remains an open question but is likely related to the saturation of $L_\theta(T)$ as discussed in the text.

On a related note, we modify the following expression in the text, ‘broadening of the QCP of the SIT’ -> ‘broadening of the SIT’. The previous expression may mislead the readers that $\zeta_{qf}(B)$ diverges at B_{MI} and B_{SM} . Our claim is that $\zeta_{qf}(B)$ diverges at QCP inside the AM state.

Comments 5:

5. This is not the first study of Nernst effect in MoGe. The vortex Nernst signal was already measured in Phys. Rev. Lett. 126, 077001 (2021), with a peak three times larger than what

was seen here. This difference may be due to the difference in the thickness of the thin films studied in the two cases. However, it should not be overlooked.

Our response:

We appreciate the valuable comments. The vortex Nernst effect of amorphous (a-)Mo_xGe_{1-x} was previously reported by C. W. Rischau *et al.* [PRL 126 077001 (2021), added as Ref. 33 in the revised manuscript] and also by our group [PRL 125, 257001 (2020), Ref. 10]. We summarize these results below. Rischau *et al.* measured a multi-layered sample consisting of alternating 10 layers of MoGe (6 nm) and insulator (4.5 nm), and the maximum amplitude of the vortex Nernst signal (= N_{\max}) is 8.3 $\mu\text{V/K}$. We measured thin film samples with a thickness of 12 nm in PRL(2020) and 10 nm in the present manuscript, and N_{\max} for both samples are $\sim 3 \mu\text{V/K}$. Thus, the value of N_{\max} we obtained are reproducible and robust. As the reviewer #1 points out, the difference of N_{\max} may be due to the difference of effective thickness or dimensionality. We add the above discussion to the revised text.

	Sample	T_{c0} (K)	N_{\max} ($\mu\text{V/K}$)
Rischau et al. , PRL (2021)	multilayer (MoGe(6 nm)/insulator(4.5 nm) $\times 10$)	6	8.3
Ienaga et al. , PRL (2020)	film (12 nm)	2.58	2.9
Ienaga et al. , the present manuscript	film (10 nm)	2.38	2.7

Comments 6:

6. I also note that the vortex α_{xy} peaks to 30 nK/A. Given the amplitude of the quantum of the magnetic flux, this corresponds to a vortex entropy of $\sim 4 k_B$ per layer. I understand that this is not an essential part of the topic discussed in this paper. However, a short discussion would be useful to the reader interested in the observation that vortex entropy per sheet is of the order of the Boltzmann constant in many superconductors (See table 1 in Physica C 591, 1353975 (2021) and table 1 in J. Phys.: Condens. Matter 35 074003(2003)).

Our response:

We appreciate the valuable suggestion. The maximum amplitude of α_{xy} (= N/R_{\square}) = 30 nA/K is converted into vortex entropy $s_{\phi} = \phi_0 \alpha_{xy} = 6.2 \times 10^{-23} \text{ J/K} = 4.5 k_B$. Since the value of s_{ϕ} represents the vortex entropy per *film thickness* (= 10 nm) in a two-dimensional system, the vortex entropy per *unit layer* ($\sim 3 \text{ \AA}$) is calculated to be $0.14 k_B$. This value is still close to k_B . In the revised text, we add the above discussion by citing Physica C 591, 1353975 (2021) and J. Phys.: Condens. Matter 35, 074003 (2023) as Refs. 51 and 52, respectively.

We again appreciate the valuable comments of the reviewer #1 that help improve our manuscript and would like to ask thoughtful consideration of this revised manuscript for publication in Nature Communications.

Reply to Reviewer #2

We would like to thank the reviewer #2 for appreciating the quality of our experiments: *‘the authors performed very high-quality and detailed experimental and analysis work,’* and for constructive comments to our manuscript. We have revised the text according to the valuable comments and suggestions.

Comments 1:

1. My major problem with this manuscript is that, despite the fact that the authors claim to present an SMIT, they do not show any insulating behavior. In Fig. 1 a,b an insulating phase is not seen. The authors claim that above “B_{MI}” the resistance shows a logarithmic-like increase with decreasing temperature (though the exact temperature dependence is not shown). Such a weak T dependence is characteristic of dirty metal behavior due to weak localization (or other phenomena) rather than an insulator that should show exponential dependence of the resistance with decreasing temperature. It is not clear how the authors define the M phase versus the I phase.

Our response:

We sincerely appreciate the important suggestions to improve the accuracy of our manuscript. As shown in Fig. 1b, resistance curves above 5.6 T show a logarithmic-like increase with decreasing temperature. The increase of resistance in the temperature range we can access is much smaller than an exponential increase observed in InO_x and TiN. As the reviewer #2 mentioned, such a behavior is regarded as a dirty metal or a “bad” metal according to Ref. 5. However, it is expected that the logarithmic behavior eventually crosses over to the exponential one by further decreasing temperature as mentioned on p.23 in Ref. 5 and by Y. Liu *et al.*, [PRB 47, 5931 (1993), added as Ref. 48 in the revised text]. Generally, the crossover temperature is inaccessibly low. Indeed, A. Yazdani *et al.* [PRL 74, 3037 (1995)] and D. Ephron *et al.* [PRL 76, 1529 (1996), Ref. 8] stated that a-Mo_xGe_{1-x} films exhibit the SMIT, where the insulator phase shows the logT dependence. In this sense, we concluded that our a-Mo_xGe_{1-x} film also shows the SMIT. In the revised manuscript, we add a detailed statement on the definition of an insulator and a dirty metal, and we use the word “insulator” to represent a regime with the logarithmic-like increase after noticing that it is also classified as a “dirty metal”.

Comments 2-1:

2. I am a bit confused about the interplay of quantum amplitude and phase fluctuations. On one hand the authors claim that the AM is due to vortex creep. On passing, I have to say that I find this claim a bit too definitive. The presence of the AM and its possible origin is very much under debate and most of the references quoted by the authors do not conclude voltage creep as the cause for AM.

Our response:

At first, we would like to ask the reviewer #2 the following typo in the comments, ‘voltage creep as the cause for AM’ -> ‘vortex creep as the cause for AM.’ As we commented in the text, “The

origin of the superconductor-metal-insulator transition (SMIT) has been debated [8–22]”, the mechanism of the metallic dissipation at $T = 0$ is still under debate. However, as we explained in the text, some of recent experimental studies have detected convincing evidence for the vortex contribution to the AM state [Refs. 10, 15, 16 in the text]. To reduce the definitive impression, we revise a sentence as the following: ‘Theories predicted that it is attributed to quantum creep of vortices’ -> ‘Some of theories predicted that it is attributed to quantum creep of vortices’

Comments 2-2:

But even if this claim was true this would mean that phase fluctuations and vortices are very important and would dominate the Nernst effect. On the other hand, in their analysis of α_{xy} and determining the limit for the quantum fluctuation regime the authors only consider amplitude fluctuations. In this respect I find the determination of the quantum fluctuation regime questionable. Since the main point of the manuscript is to determine the broadened quantum critical phase of the AM it is important to identify all sources for fluctuations.

Our response:

We are concerned that our writing may have led to confusion about our experimental detection of the quantum fluctuations of the phase. We referred to the quantum fluctuations of the phase as the quantum vortex liquid (QVL) in the text. We mentioned on page 5, line 144 in the previous manuscript (page 6, line 160 in the present manuscript), ‘*Below the $B_{c2}(T)$ line in the thermal vortex-liquid phase, N and α_{xy} have large magnitudes, which are due to vortex motion. They are observed down to the lowest temperature 0.1 K in the high- B range of the AM state, indicating that the AM state originates from the vortex liquid due to the quantum fluctuations, namely quantum vortex liquid (QVL), as reported in our previous work [10].’ and on page 6, line 177 in the previous manuscript (page 7, line 194 in the present manuscript), ‘*Thus, $B_{MI} (\approx B_{c2}(0))$ is not a QCP but a crossover point, below which the quantum amplitude fluctuations is transformed into the QVL.*’ As represented in these sentences, we have detected the quantum fluctuations of the phase as the QVL. Indeed, for the Nernst effect due to the mobile vortices, no available theory has been reported to distinguish the thermal and quantum regimes. This is in sharp contrast to the Nernst effect due to the amplitude fluctuations, for which theories [Refs. 41-44] provide a prescription for detecting the thermal-to-quantum crossover line $T^*(B)$. Therefore, experimental confirmation of the QVL is given by the observation that the vortex Nernst effect survives even at low temperatures where the temperature dependence of $B_{c2}(T)$ is almost saturated and thermal effect is well suppressed. Furthermore, as shown in Figs. 2a and 2b, the thermal-to-quantum crossover line $T^*(B)$, which is detected as a peak position in the temperature dependence of $\alpha_{xy}(T)$, is observed not only in the region of the amplitude fluctuations above the $B_{c2}(T)$ line but also in the vortex-liquid state below the $B_{c2}(T)$ line. This result also suggests the existence of the QVL below $T^*(B)$.*

Comments 3:

3. Another importance of this work raised by the authors in the conclusion is that the work shows the usefulness of the Nernst effect to study QPTs in superconducting systems. It should

be noted that reference 36 showed a detailed study of quantum criticality of the SIT using Nernst effect, though in that case there was no intermediate AM.

Our response:

We appreciate the helpful comment. As pointed out by the reviewer #2, A. Roy *et al.* [PRL 121, 047003 (2018), Ref. 38] found quantum criticality of the SIT using a Nernst effect. Thus, we revise our text by citing Ref. 38 in the introduction as an important work.

Moreover, we add another important work as Ref. 56 in the discussion, which reports the quantum criticality of the SIT using a specific heat measurement.

Comments 4-1:

4. There are a few small typos in the text for example:

(i) In the middle of page 5 “ \hbar ” should be the Plank constant and not the Dirac constant.

Our response:

We correct the name of \hbar from “the Dirac constant” to “the reduced Planck constant”. Although in some literature \hbar is called the Dirac constant, we have appreciated that the reduced Planck constant is more familiar.

Comments 4-2:

(ii). Caption of Fig. 1 a rather than “fixed B” it should be “different B”.

Our response:

We correct the caption of Fig. 1a and other similar phrase in the text and captions.

We again appreciate the valuable comments of the reviewer #2 that help improve our manuscript. We would like to ask thoughtful consideration of this revised manuscript for publication in Nature Communications.

Summary of changes (Pages and lines correspond to those in the revised manuscript)

***Changed parts are highlighted in the pdf version of the revised manuscript.**

• **Main text**

Page 2, line 32: ‘Theories predicted that’ → ‘Some of theories predicted that’

Page 2, line 36: ‘broadening of the QCP of the SIT’ → ‘broadening of the SIT’

Page 3, line 61: ‘a 2D film of amorphous (a-)Mo_xGe_{1-x},’
→ ‘a 2D film of amorphous (a-)Mo_xGe_{1-x} with a thickness of 10 nm’

Page 3, line 66:

‘the Nernst effect is a very useful method to detect a QCP in superconducting systems’
→ ‘the Nernst effect is a very useful method to detect a QCP in superconducting systems as reported for the SIT [38]’

Page 3, line 68: ‘sheet resistance $R_{\square}(T)$ in fixed B ’ → ‘sheet resistance $R_{\square}(T)$ in different B ’

Page 4, line 92: the following sentences are added:

‘As mentioned in our previous report [10], we regard the weak localization behavior as a character of an insulator [6, 10, 11], which is expected to cross over to a strongly insulating state with exponential divergence at much lower temperatures [5, 48]. The weakly localized state is also identified as a dirty metal caused by quantum correction [5].’

Page 4, line 97: ‘at fixed T from 0.1 K to 2.4 K’ → ‘at different T from 0.1 K to 2.4 K’

Page 4, line 103: the following sentences are added:

‘The maximum amplitudes of the vortex Nernst signal N_{\max} in the B - T range studied is 2.7 $\mu\text{V/K}$ at 1.2 K in 2.8 T. This value is comparable to $N_{\max} = 2.9 \mu\text{V/K}$ of the 12 nm-thick a-Mo_xGe_{1-x} film with $T_{c0} = 2.58$ K used in our previous work [10] but smaller than $N_{\max} = 8.3 \mu\text{V/K}$ of a multi-layered a-Mo_xGe_{1-x} film with $T_{c0} \sim 6$ K [33]. The difference of N_{\max} may be due to the difference of effective thickness or dimensionality.’

Page 5, line 120: the following sentences are added:

‘Recent studies of the vortex Nernst effect have suggested that a maximum value of the vortex transport entropy per unit layer s_{ϕ}^{sheet} is of the order of k_B in many superconductors [33, 51, 52]. In our experiment, a maximum amplitude of $\alpha_x^{\phi} = 30$ nA/K at 1.6 K in 0.8 T is converted into $s_{\phi} = \phi_0 \alpha_{xy}^{\phi} = 4.5 k_B$. Since s_{ϕ} represents the vortex transport entropy per film thickness (= 10 nm), s_{ϕ}^{sheet} is calculated to be 0.14 k_B supposing a unit layer thickness of ~ 3 Å for a-Mo_xGe_{1-x}. This value is still close to k_B .’

Page 5, line 136: ‘at fixed T above $T_{c0,R}$.’ → ‘at different T above $T_{c0,R}$.’

Page 5, line 143: ‘where \hbar is the Dirac constant’ → ‘where \hbar is the reduced Planck constant’

Page 6, line 149:

‘we can determine $T_{c0} = 2.38$ K (see Supplementary Fig. S3), whose value is quite close to $T_{c0,R} = 2.36$ K.’

→ ‘we can determine $T_{c0} = \underline{2.36}$ K (see Supplementary Section III-1), which coincides with $T_{c0,R} = \underline{2.36}$ K.’

Page 6, line 158: the following sentences are added:

‘The existence of $B^*(T)$ in the normal state is also confirmed by scaling analysis for α_{xy}/B (see Supplementary Section III-2).’

Page 6, line 168: ‘for fixed B from 4.6 T to 10 T’ → ‘for different B from 4.6 T to 10 T’

Page 7, line 180: the following sentences are added:

‘(see Supplementary Section III-3)’

Page 7, line 184: ‘(see Supplementary Eq. 2)’ → ‘(see Supplementary Section III-3)’

Page 7, line 195: ‘the quantum amplitude fluctuations is transformed’ → ‘the quantum amplitude fluctuations are transformed’

Page 7, line 197: the following sentences are added:

‘The existence of $T^*(B)$ is also confirmed by scaling analysis for α_{xy}/T (see Supplementary Section III-4).’

Page 7, line 201: ‘ $T_{c0} = 2.38$ K’ → ‘ $T_{c0} = \underline{2.36}$ K’

Page 8, line 215: ‘(see also Supplementary Fig.S2)’ → ‘(see also Supplementary Section II)’

Page 8, line 221: ‘(see Supplementary Fig. S3)’ → ‘(see Supplementary Section III-3)’

Page 8, line 238: the following sentences are added:

‘(see Supplementary Section III-4)’

• **References(Main text)**

The following are additionally cited:

[27] Pourret, A., Aubin, H., Lesueur, J., Marrache-Kikuchi, C. A., Berg'e, L., Dumoulin, L., and Behnia, K. Length scale for the superconducting Nernst signal above T_c in Nb_{0.15}Si_{0.85}. Phys. Rev. B 76, 214504 (2007)

[33] Rischau, C. W., Li, Y., Fauqu'e, B., Inoue, H., Kim, M., Bell, C., Hwang, H. Y., Kapitulnik, A., and Behnia, K., Universal bound to the amplitude of the vortex Nernst signal in superconductors. Phys. Rev. Lett. 126, 077001 (2021).

- [48] Liu, Y., Haviland, D. B., Nease, B., and Goldman, A. M. Insulator-to-superconductor transition in ultrathin films. *Phys. Rev. B* 47, 5931 (1993).
- [51] Huebener, R. P., and Ri, H.-C. Vortex transport entropy in cuprate superconductors and Boltzmann constant. *Physica C* 591, 1353975 (2021).
- [52] Behnia, K. Nernst response, viscosity and mobile entropy in vortex liquids. *J. Phys.: Condens. Matter* 35, 074003 (2023).
- [53] Kapitulnik, A., Palevski, A., and Deutscher, G. Inhomogeneity effects on the magnetoresistance and the ghost critical field above T_c in thin mixture films of In-Ge. *J. Phys. C: Solid State Phys.* 18, 1305-1312 (1985).
- [56] Poran, S., Nguyen-Duc, T., Auerbach, A., Dupuis, N., Frydman, A., and Bourgeois, O. Quantum criticality at the superconductor-insulator transition revealed by specific heat measurements. *Nature Commun.* 8, 14464 (2017)

• **Figures(Main text)**

Fig. 2a-d: T_{c0} is changed from 2.38 K to 2.36 K and other changes were made accordingly.

Fig. 2a-c: the plot of T^* (blue squares) in 4.6 T is deleted.

Fig. 2a: the value represented by the color scale bar is changed from $\log(N [\mu\text{V/K}])$ to $\underline{N (\mu\text{V/K})}$.

Fig. 2b: the value represented by the color scale bar is changed from $\log(\alpha_{xy} [\text{nA/K}])$ to $\underline{\alpha_{xy} (\text{nA/K})}$

Fig. 2d: the value represented by the color scale bar is changed from $\log(\alpha_{xy}^{\text{fl, theo}} [\text{nA/K}])$ to $\underline{\alpha_{xy}^{\text{fl, theo}} (\text{nA/K})}$.

Caption of Fig. 2:

‘the input parameters are only $T_{c0} = 2.38$ K and $B_{c2}(0) = 5.5$ T’ → ‘the input parameters are only $T_{c0} = \underline{2.36}$ K and $B_{c2}(0) = 5.5$ T’

Caption of Fig. 2: the following sentences are added:

‘Error bars for B^* and T^* represent ranges of B and T in which $\alpha_{xy}(B)$ at fixed T and $\alpha_{xy}(T)$ at fixed B exceed 95 % of their peak amplitudes, respectively.’

‘using the codes provided in Ref. 43.’

Fig. 3: the data of $\alpha_{xy}(T)$ in 4.6 T are deleted.

Caption of Fig. 3: ‘for fixed B from 4.6 T to 10 T’ → ‘for different B from 4.6 T to 10 T’

Fig. 4:

the blue line representing $T^*(B)$ is added and the color arrangement is changed.

Caption of Fig. 4:

‘broadening of the QCP of the SIT’ → ‘broadening of the SIT’
‘which is verified by finding B_{QCP} (solid orange circle) in this work.’ → ‘which is verified in this work by finding B_{QCP} (solid orange circle) as a field that the experimental $T^*(B)$ line approaches as indicated by a solid blue line.’

• **Supplemental Information**

The section numbers are changed from 1, 2, 3, 4 to I, II, III, IV.

Page 4, line 38:

‘3. Thermal and quantum critical behavior in $\alpha_{xy}(T, B)$ ’

→ ‘III. Critical behaviors and correlation lengths in $\alpha_{xy}(T, B)$ ’

Page 4, line 39: subsection title is added as ‘1. Thermal critical behavior in $\alpha_{xy}(T, B)$ ’.

Page 4, line 42:

‘ $\alpha_{xy}^{\text{fl}} = (k_{\text{B}}e^2/6\pi\hbar^2)\zeta_{\text{GL}}^2 B \propto B/\ln(T/T_{\text{c0}})$ ’ → ‘ $\alpha_{xy}^{\text{fl}}/B = (k_{\text{B}}e^2/6\pi\hbar^2)\zeta_{\text{GL}}^2 \propto \zeta_{\text{GL}}^2$ (1)’

Page 4, line 43: ‘the Dirac constant’ → ‘the reduced Planck constant’

Page 4, line 44:

‘This means that an initial slope of’

→ ‘This means that α_{xy}/B is independent of B in the low- B region ($B \ll B^*$) and an initial slope of’

Page 4, line 47: ‘this equation is modified’ → ‘Eq. (1) is modified in the low- B region ($B \ll B^*$)’

Page 4, line 47:

‘ $\alpha_{xy}^{\text{fl}} = (e^2 D/6\pi^2 \hbar)(B/T \ln(T/T_{\text{c0}}))$, (1)’ → ‘ $\alpha_{xy}^{\text{fl}}/B = (e^2 D/6\pi^2 \hbar)(1/T \ln(T/T_{\text{c0}})) \propto \zeta_{\text{GL}}^2/T$, (2)’

Page 4, line 51:

‘As seen from the B dependence of α_{xy} at fixed $T (> T_{\text{c0}})$ shown in Fig. 1i, where $\alpha_{xy}(B)$ is attributed to $\alpha_{xy}^{\text{fl}}(B)$, $\alpha_{xy}/B|_{B \rightarrow 0}$ increases with decreasing T .’

→ ‘In Fig. S3a, we display the B dependence of α_{xy}/B at different T , which is converted from Fig. 1g,i. Above $T_{\text{c0}} (= 2.36 \text{ K})$, where $\alpha_{xy}(B)$ is attributed to $\alpha_{xy}^{\text{fl}}(B)$, α_{xy}/B exhibits saturation to a constant value in the low- B region, giving the value of $\alpha_{xy}/B|_{B \rightarrow 0}$.’

Page 4, line 53: ‘In Fig. S3a’ → ‘In Fig. S3c’

Page 4, line 54: ‘extracted from Fig. 1i’ → ‘extracted from Fig. S3a’

Page 4, line 56: ‘obtained with $T_{\text{c0}} = 2.38 \text{ K}$.’ → ‘obtained with $T_{\text{c0}} = \underline{2.36 \text{ K}}$.’

Page 5, line 72:

new section (III-2) is added to discuss the existence of the ghost critical field $B^*(T)$.

Page 6, line 82: subsection title is added as ‘3. Quantum critical behavior in $\alpha_{xy}(T, B)$ ’.

Page 6, line 84:

‘the theoretical value of α_{xy}^{fl} at $T \ll T^*$ ’ \rightarrow ‘the theoretical value of α_{xy}^{fl} in the low- T region ($T \ll T^*$) for fixed B ’

Page 6, line 86:

‘The exact expression’ \rightarrow ‘The exact theoretical expression’

Page 6, line 86:

‘ $\alpha_{xy}^{\text{fl}} = (k_B^2/12\hbar D)(T/B \ln(B/B_{\text{QCP}}))$.’ \rightarrow ‘ $\alpha_{xy}^{\text{fl}}/T = (k_B^2/12\hbar D)(1/B \ln(B/B_{c2}(0))) \propto \xi_{\text{qf}}^2/B$. (5)’

Page 6, line 87: the following sentences are added below Eq. (5):

‘This is a quantum counterpart of Eq. (2). In the present case of $B_{\text{QCP}} \neq B_{c2}(0)$, ξ_{qf} should be rewritten as $\xi_{\text{qf}} = \xi_0/\sqrt{\ln(B/B_{\text{QCP}})}$ and the variable B in Eq. (5) should be replaced by $B(B_{c2}(0)/B_{\text{QCP}})$ ($\equiv B' = B(5.5/4.6)$) as deduced from the discussion in the section II. Thus, we obtain $\alpha_{xy}^{\text{fl}}/T = (k_B^2/12\hbar D)(1/B' \ln(B'/B_{c2}(0))) \propto \xi_{\text{qf}}^2/B'$. (6)’

Page 6, line 93: the following sentences are added:

‘In Fig. S3d, we show the T dependence of α_{xy}/T at different B above $B_{c2}(0)$ (= 5.5 T), which is converted from Fig. 3. Because of the sensitivity limit of Nernst signals N , α_{xy}/T below $B_{N \rightarrow +0}(T)$ is plotted. α_{xy}/T exhibits a trend to be saturated to a constant value in the low- T region. Note that below $B_{c2}(0)$, α_{xy}/T does not show the saturation behavior but a divergent behavior caused by quantum criticality as discussed in the section IV.’

Page 6, line 97: ‘In Fig. S3b’ \rightarrow ‘In Fig. S3f’

Page 6, line 98: ‘from Fig. 3’ \rightarrow ‘from Fig. S3d’

Page 6, line 100: ‘in Fig. S3b’ \rightarrow ‘in Fig. S3f’

Page 6, line 105:

new section (III-4) is added to discuss the existence of the ghost temperature $T^*(B)$.

• **References(Supplemental Information)**

The following are additionally cited:

[10] Pourret, A., Aubin, H., Lesueur, J., Marrache-Kikuchi, C. A., Berg'e, L., Dumoulin, L., and Behnia, K., Length scale for the superconducting Nernst signal above T_c in Nb_{0.15}Si_{0.85}. Phys. Rev. B 76, 214504 (2007).

[14] Sondhi, S. L., Girvin, S. M., Carini, J., and Shahar, D., Continuous quantum phase transitions. Rev. Mod. Phys. 69, 315-333 (1997).

• **Figures(Supplemental Information)**

Caption of Fig. S1: 'at fixed T ' → 'at different T '

Caption of Fig. S2: the following sentences are added:

'Error bars for B^* represent B ranges in which $\alpha_{xy}(B)$ at fixed T exceeds 95 % of the peak amplitude.'

Fig. S3a,b,d,e: new figures are added in the revised version.

Fig. S3c,f: they are revised from Fig. 3a,b in the previous version.

Caption of Fig. S3: the sentences are revised to explain the new figures.

REVIEWERS' COMMENTS

Reviewer #1 (Remarks to the Author):

The authors have fully addressed my concerns and I recommend the publication of this work, which is a detailed study of the Nernst effect in a disordered superconductor, providing new input for the ongoing debate on the origin of anomalous metallicity.

Reviewer #2 (Remarks to the Author):

The authors have answered all of my questions and revised the manuscript according to the comments of Reviewer 1 and me. Though I still don't entirely agree with the authors on the defining the region of logarithmic dependence of resistance on temperature as an "insulator", I don't think this should prevent publication in Nature communications. Therefore, I recommend accepting this manuscript for publication.